# Blood Lead Monitoring in a Former Mining Area in Euskirchen, Germany—Volunteers across the Entire Population

**DOI:** 10.3390/ijerph19106083

**Published:** 2022-05-17

**Authors:** Jens Bertram, Christian Ramolla, André Esser, Thomas Schettgen, Nina Fohn, Thomas Kraus

**Affiliations:** 1Institute for Occupational, Social and Environmental Medicine, University Hospital Aachen, 52074 Aachen, Germany; anesser@ukaachen.de (A.E.); tschettgen@ukaachen.de (T.S.); nfohn@ukaachen.de (N.F.); tkraus@ukaachen.de (T.K.); 2Public Health Department, District of Euskirchen, 53879 Euskirchen, Germany; christian.ramolla@kreis-euskirchen.de

**Keywords:** BLL, former, lead, mining area, blood, Germany

## Abstract

After centuries of mining in the district of Euskirchen, that is, in the communities of Mechernich and Kall, the lead concentration in the soil remains high, often exceeding regulatory guidelines. To clarify the lead body burden among residents in the region, a human biomonitoring study on a voluntary basis was initiated in which the blood lead level (BLL) was assessed. A questionnaire was distributed to evaluate lead exposure routes and confounders. Overall, 506 volunteers participated in the study, of whom 7.5% were children and adolescents, 71.9% were adults from 18 to 69 years, and 19.4% were residents 70 years or older. While the BLLs in the adult population were inconspicuous, among the children and adolescents investigated, 16.7% of the children between 3 and 17 years had BLLs above the recently revised German reference values for BLL in children. These results point towards a higher lead exposure in children living in the region. The hierarchical regression analysis based on the BLL and the questionnaire revealed the significant influence of the factors age, sex, smoking, construction age of the real estate, occupancy, and intensive contact with soil on the BLL. Measures to reduce lead exposure include a focus on improved personal and domestic hygiene to minimize lead intake.

## 1. Introduction

Lead has been identified as a hazardous substance since antiquity. A wide variety of adverse effects that lead can have on systems or organs in the human body have been demonstrated [1]. The most difficult to estimate are the neurotoxic effects, which affect the peripheric and central nervous system. Lead was proven to already adversely affect the cognitive development of children with BLL below 50 µg/L. Additionally, children resorb about five times more lead than adults when ingested orally, that is, up to 50% of the lead ingested. Lead binds primarily to hemoglobin and is therefore easily distributed throughout the body. Excretion takes place via urine and feces. The half-life time of lead in blood is about 30 days. The BLL reflects the exposure of the previous months. Bones and teeth are depots for long-term storage, with approximately 94 and 76% of the body burden deposited in the bones of adults or children, respectively, thus, representing the cumulative long-term exposure. To date, there is no BLL that is considered safe [2,3,4,5,6,7,8].

In the 20th century, the main applications of lead were in water pipes, in indoor and outdoor paints, and of special significance, as an anti-knock agent in fuels until the use of lead was phased out in the 1970s, contributing to the BLL of the general population. Today, the most important field of use for lead is in the battery technology sector. However, active or former processing and mining sites still pose a source of concern for the local citizenship [9,10,11,12].

Since the burden of the local population differs widely depending on the region and the mining site [13,14,15,16], the question that arose was if the local population in Mechernich and Kall shows higher BLLs than the general German population.

New appraisals of the toxicity of lead were introduced in Germany recently. Lead has formerly (since 2007) been categorized as a class 2 carcinogenic substance, that is, as car-cinogenic for animals by the German Research Foundation. In 2021 it was reevaluated and classified as a category 4 substance, that is, as a substance with a non-carcinogenic mechanism [17,18].

Over the recent decades ongoing efforts to reduce lead in the environment have been undertaken worldwide. Measures such as the phasing out of leaded fuels, the ban of lead-containing paints, and a reduction of lead in toys have resulted in a lower lead bur-den within the general population in countries around the globe [10,19,20,21,22]. Therefore, the according reference values (based on the 95th percentile of the general population) have been adapted to the new situation. The German reference values derived for adult males and females decreased from 90 µg/L and 70 µg/L for men or women, respectively, to 40 µg/L and 30 µg/L for men or women in 2018. In 2019, the reference value of 35 µg/L for lead in the blood for children from 3 to 14 years was adjusted to 20 µg/L for 3 to 10-year-old boys and to 15 µg/L for 11 to 17-year-old boys as well as for 3 to 17-year-old girls [7,23,24,25].

In the former mining area of Mechernich and Kall in North Rhine-Westphalia, the mining of lead ore took place for centuries. The mining activities came to an end in 1957, but lead prevails in the soil of the region. The analysis of soil samples in 1986 demonstrated lead soil concentrations of up to more than 10,000 mg/kg dry mass throughout large areas in the respective communities [26]. Following German regulations on the presence of lead in the soil of housing areas, lead values in the soil must not exceed 400 mg/kg dry mass. The soil designated for new construction sites in the area showed elevated lead soil concentrations, resulting in an interest in the assessment of the body lead burdens of the residents of the communities.

As a consequence, the communities decided to offer a voluntary blood lead analysis for all the residents of both communities, Mechernich and Kall, to evaluate the current lead burden of the local population compared to the revised RVs in order to clarify whether or not higher than expected BLLs, as known from the literature, that is, from the German RVs, prevail in the region. A questionnaire was provided to assess further information regarding the personal and domestic characteristics and to identify further potential sources.

It has to be stated that the study at hand does not claim to pose a representative random sample since it is only based on volunteer residents.

BLLs in adults did not differ from the background burden of the general population. Indications of a higher-than-expected body burden in children and minors were found within the limitations of a sample group being too small to be statistically robust.

Among commonly known factors contributing to the BLL, occupancy, intense soil contact, as well as the age of the house, were identified as having a significant effect on BLLs. A general effect of the climate on BLLs is hypothesized in the discussion section and might give way for future studies dealing with the issue of metal body burden from min-ing activities from a different angle.

## 2. Materials and Methods

### 2.1. Sampling

The summer months were selected to investigate the near-maximum BLLs in the local population due to the effect of seasonal changes on lead in the environment, which peak in the summer months. This phenomenon can be explained by the augmented inhalation of dust, the consumption of home-grown foods, and a prolonged time spent outside, thus maximizing the exposure in the warm period of the year [4,27,28]. The investigation should reflect a worst-case scenario of the lead burden in the local population. The study was announced publicly after several local meetings. The sampling was performed on the 25th of June and the 4th of July 2019 (s. Figure 1). The BLL reflects the individual exposure of the preceding weeks and months due to the half-life time of lead in blood. A sampling time after the summer holidays, which tend to be spent elsewhere, was not considered ideal to evaluate the local exposure; thus, the sampling was performed before the summer holidays.

The samples were collected from trained personnel of the local hospital in Mechernich, supervised and organized by the health department of Euskirchen. Written informed consent was obtained from the participants or their legal representatives. Blood samples were collected in 4.9 mL EDTA Monovettes (Sarstedt, Nümbrecht, Germany) and stored in a fridge overnight. The samples were transported by courier to the Institute for Occupational, Social and Environmental Medicine Aachen (IASU) of the university hospital of RWTH Aachen for analysis.

### 2.2. Questionnaire

A questionnaire was handed out covering the topics age, sex, occupancy, age of the residence, garden ownership, time spent in the garden, intense soil contact, consumption frequency of home-grown vegetables and fruits, type of vegetables and fruits consumed, potential private freshwater wells, consumption of bowels, smoking habits, alcohol consumption, lead-related hobbies such as membership in a gun club, illness, medicaments, and in the case of small children, diet and hand-to-mouth contact. The residency was categorized into one of four regions. The areas were classified by differing lead soil concentrations in which Region 1 had lead soil concentrations above 5000 mg/kg, and Region 2 had concentrations from 1000 to 5000 mg/kg. Region 3 comprised regions near newly assigned real estate developments. Region 4 did not form part of the other regions covered, that is, with unknown lead soil concentrations within the communities (s. Appendix A).

### 2.3. Analysis

The analysis was performed using an Agilent 8900 inductively-coupled plasma triple quadrupole mass spectrometer system (Agilent, Darmstadt, Germany). The samples were diluted 1:50 with Millipore water, acidified with 20 µL of 65% nitric acid, and mixed with 300 µL of a 10% Triton X solution. 100 µL Rhodium (1 mg/L) was used as internal standard. The analysis was performed in the single quadrupole Helium mode. The limit of detection was 0.01 µg/L. The limit of quantification was 0.03 µg/L. The calibration curve ranged from 20 to 7000 ng/L using doted blood samples and blank solutions.

Certified reference material was used in all of the analytical runs for quality assurance every 12 samples. The laboratory regularly and successfully participates in the external quality control scheme of the Erlangen-Nuremberg University in both environmental as well as occupational sample materials.

### 2.4. Statistics

All statistical analyses were performed using an SPSS 25 (IBM (2017) SPSS Statistics for Windows, Version 25, Armonk, NY, USA). Hierarchical linear regression was performed to quantify the influence of the different living factors on BLLs. The distribution of the BLLs was left-censored and right-skewed. Therefore, the normal distribution of BLL residuals was investigated by Q–Q plots.

## 3. Results

### 3.1. Sample Collective

Overall, 506 volunteers participated in the study, of which 38 (7.5%) were children and adolescents from 2 to 17 years, 364 (72.9%) were adults from 18 to 69 years, and 98 (19.4%) were adults of 70 years or older.

Three participants did not provide a questionnaire, and another three did not state their age. Overall, 222 (43.9%) of the participants were male, 281 (55.5%) were female, and three were of unknown sex due to the three lacking questionnaires (Figure 2). Significantly more participants were females (*p* ≤ 0.010).

Of the regions defined, Region 1 comprised N = 86, Region 2 N = 227, Region 3 N = 46, and Region 4 comprised N = 142 participants. The residency of five participants was unknown.

### 3.2. BLL

Six out of thirty-six children and minors between 3 and 17 years (16.7%) had a BLL above the 95th percentile of the general population (3 to 10 -year-old boys 20 µg/L, 11 to 17-year-old boys, and 3 to 17-year-old girls 15 µg/L). The mean BLLs in our investigation for 3 to 10-year-old boys was 14.6 µg/L. 3 to 17-year-old girls had a mean BLL of 10.6 µg/L, and 11 to 17-year-old boys had a mean BLL of 10.8 µg/L. The maximum BLLs measured were 25.4 µg/L in a 5-year-old girl and 26.6 µg/L in an 11-year-old boy (Table 1).

The percentage of adult participants that exceeded the reference values was 6.5 % (n = 14) for 18 to 69-year-old women (30 µg/L) and 6.1 % (n = 9) for 18 to 69-year-old men (40 µg/L). The mean values were 15.4 µg/L for 18 to 69-year-old women and 17.8 µg/L for 18 to 69-year-old men, respectively. The maximum values observed were 69.6 µg/L in a 59-year-old man who had professional contact with soldering materials and 56.0 µg/L in a 57-year-old woman.

The mean levels in the elderly adult population were higher compared to the 18 to 69-year-olds. Women had mean values of 22.2 µg/L compared to 15.4 µg/L in younger adult women, while men had mean values of 24.4 µg/L compared to 17.8 µg/L in younger adult men. The maximum values observed were 54.8 µg/L in a 72-year-old woman and 55.7 µg/L in a 92-year-old man. The BLLs depending on age measured in decades, is given in Figure 3. In the age groups of the 20 to 29-year-olds and the 30 to 39-year-olds, differences between both sexes can be observed. These variances might be explained by the five individuals who showed marked RV exceedances in comparatively small subgroups (s. Figure 2), thus having a major effect on the mean BLL. Three of these five reported intense soil contact, while one did not make a statement. No professional contact was stated.

### 3.3. Statistics and Questionnaires

In a hierarchical regression analysis, we examined the influence of the life circumstances recorded in the questionnaire, including the age, sex, and housing situation. The biggest influence was revealed for age and the duration of the residency. The categorical variables occupancy region, pregnancy, and intake into the bowels did not reach the limit for significance; the details are shown in Table 2.

The number of smokers in this study was 68 (13.4%), 432 (85.4%) were non-smokers, and six persons did not make a statement.

Overall, 196 participants reported to have intense soil contact, and 266 did not report having this contact (Figure 4).

The age of the real estate is given as a boxplot in Figure 5.

## 4. Discussion

### 4.1. Sample Collective

The age distribution of the volunteers is a major limitation of the study when it comes to deriving general insights. The small number of children participating in the study prevents a statistically robust conclusion. Another drawback of the study is the unequal age distribution in the adult groups. As visible in Figure 1, the majority of the participants were between 40 and 80 years while the participants aged 18 to 40 years were in short supply. On the other hand, the age distribution between male and female participants was similar (*p* = 0.307). It remains unclear whether or not this reflects the demographics of the local population.

The study design was not intended to be statistically representative. However, random sampling of the size of the study at hand does provide valuable information on the blood lead body burden of the population in the region within the beforementioned limitations.

Due to the study design and the pseudonymization process, the data about the residence of the different participants within the communities of Mechernich and Kall and the distribution between the communities of Mechernich and Kall are unknown. The number of participants from the different regions varied and was difficult to rate. Again, it remains unclear how this distribution reflects the local population.

The underlying reasons were matters of data protection and avoiding the introduction of psychological hurdles, possibly hindering study participation. However, it was assumed that the vast majority of the volunteer participants were long-term German citizens, and a small fraction of participants with different nationalities were assumed to be too small and heterogeneous to draw conclusions from the data at hand.

### 4.2. BLL

#### 4.2.1. Minors

Since the environmental BLL-RV were derived from the 95th percentile of the general population, thus 5% of RV-exceedances were expected for the corresponding age groups depending on sex. The underage girls and boys investigated indicated more reference value exceedances than expected, which raises a cause for concern. The maximum RV exceedances in girls and boys were up to twice as high as the according RVs (s. Table 1).

Since adverse effects on the development of the neurological system have already been reported to occur below 50 µg/L a RV exceedance should be avoided, and the BLL kept as low as possible. All children with RV exceedances were therefore invited to a counseling interview, and a second blood sampling was offered after several weeks to check the development of the BLL status. However, due to the small number of subjects investigated, these results are not statistically robust.

#### 4.2.2. Adults

The number of RV exceedances observed among the adult volunteers was slightly above the 5% of exceedances expected from the general population. Two of the male adult participants were identified to have an additional exposure to lead due to the professional handling of arms and ammunition and dealing with lead-containing solder materials.

In other words, a small number of individual RV exceedances was observed. Since the study was based on volunteers, the small exceedance of the expected percentage of BLLs above the RV and the maximum values observed did not indicate a general exceedance of the 95th percentile. The number of exceedances did not give a reason for concern.

In the adult population, the maximum concentrations for both sexes observed were nearly twice as high as the corresponding environmental reference values (Table 1).

However, these observed maximum values were considerably lower than the German Biological Tolerance Value for Occupational Exposures (BAT) of 150 µg/L, which is valid for employees and workers. Per definition, the BAT value is not considered to cause adverse health effects, even after repeated exposure to lead, resulting in BLLs below the BAT during work-life.

The interpretation of the results of the participants who are 70 years or older is difficult due to lacking the RV of the age group beyond 70. Underlying the RV of the adult groups from 18 to 69, the number of reference exceedances would indicate a significant percentage of cases (16.3% each for men and women) exceeding the RV in the same magnitude as observed in the underaged group. However, since no RV exists, the interpretation of the data is difficult. The mean levels in the elderly participants were higher compared to the 18 to 69-year-olds, probably indicating a tendency toward a higher body burden. These findings can be explained by the increased lead released from decreasing bone volume, which occurs as part of the aging process in adults, most likely resembling the overall, ten-fold higher lead burden during the decades when leaded fuels were used. To the authors’ knowledge, other special activities, such as the number of farming activities in the region, have not changed much in the area since the closure of the mine in 1957 and therefore did not explain the trend towards higher BLL in elderly people.

### 4.3. Hierarchical Regression Analysis

Since the RVs are given depending on age groups, unsurprisingly, the factor age turned out to be correlated with higher BLLs (β = 0.218, *p* = 0.000). The differences between BLL in adults and children have been thoroughly described in the literature. A trend towards higher BLLs in older people is described. It is also known, that in younger children, higher BLLs can be observed than in older children due to age-related behavior. For instance, hand-to-mouth contact and mouthing of objects [29,30]. These findings were confirmed as well, although mean BLLs varied in the age groups between 20 to 40-year-olds, presumably due to the smaller sample size in this study compared to the cited literature.

Occupancy of the real estate was correlated with higher BLLs (β = 0.156, *p* = 0.008). It can be argued that the longer the time spent in an area with contaminated soil, the higher the possibility of absorbing lead via the inhalation of dust or ingestion, resulting in an overall higher lead body burden.

Smoking is also a long-known factor that contributes to BLLs. Among a grand variety of other substances, lead is also absorbed via the inhalation pathway. Unfortunately, the amount of tobacco consumed was not recorded. However, smoking was correlated with BLLs (β = 0.024, *p* = 0.0009), likely due to an overall tendency of the reduced smoking habits of current smokers when compared to previous decades.

With regard to sex, it is known that BLLs in males are higher than in females, which is linked to the binding of lead to hemoglobin, which has a higher concentration in the blood of males than in females [29]. Unsurprisingly, these observations have been confirmed by our study as well. The regression analysis revealed a β of 0.107 % (*p* = 0.024) for the sex of the subject.

Intense soil contact was also correlated with the participants’ BLLs (β = 0.104, *p* = 0.035). In rural areas in the region, access to a garden as part of the property is common. Overall, 266 participants reported having intense soil contact, that is, having direct contact while either working in the garden or having occupational contact with soil or playing on a playground, while 196 did not report having intense soil contact. The remaining participants did not make a statement. The inhalation of dust and the ingestion of particles while working or drinking seems to be the most plausible contribution to the BLL. A Kruskal–Wallis test revealed a significant difference in BLLs between the participants with and without intense soil contact (H = 25.7; *p* < 0.001) (Figure 3).

The age of the residency was negatively correlated with BLLs. In buildings constructed before 1973, lead plumbing and lead-containing paints might persist and contribute to non-soil-related lead exposure. A Kruskal–Wallis test showed that residents of houses built after the year 2000 (group 3) have significantly lower BLLs than residents of both groups in older houses (group 2: H 4.8, *p* < 0.01 and group 1: H = 5.9, *p* < 0.001). The difference in the BLLs of residents between house age groups 1 and 2 was not significant (see Figure 4). Again, the results fit the findings in the literature. The more recent that the house was constructed, the lower the participants’ BLL.

Time spent in the garden, region, pregnancy, alcohol consumption, and the consumption of home-grown foods and other parameters evaluated had no significant effect on the BLL. The garden time spent was close to significance, which should be considered when discussing exposure-reduction strategies.

### 4.4. Comparison to Other Studies

A study conducted by Einbrodt in 1982 investigated the BLLs of N = 1631 4 to 14-year-old children in the region [31] and found an overall higher mean BLL compared to the recent data, which can be explained by higher BLLs in the general population back in the 1980s, mainly originating from the use of leaded fuels, but also lead-containing materials, such as paints and pipes [10]. Einbrodt found statistically significant higher BLLs in some regions compared to data from children of two reference regions farther away in different parts of North Rhine-Westphalia.

The updated German RVs for children between 3 and 17 years, presented in Vogel et al., are in a similar magnitude as the ones measured in the study at hand [25]. The mean value observed in samples collected from 2015 to 2017 was 9.4 µg/L with the 95th percentile of 19.9 µg/L for both sexes in the work of Vogel et al. (differentiated to the official RV depicted in Table 1). The total mean BLL for 3 to 17-year-old minors in the study at hand was 12.0 µg/L and the 95th percentile 23.9 µg/L, which is slightly higher than in the general German population. Again, it has to be stated that these data lack robustness due to the small sample size of our study.

Various scientific literature dealing with mining sites across the world exists. For comparison with our data, some of this work is discussed in the following paragraphs.

#### 4.4.1. Children Living in the Vicinity of Mining Sites

Table 3 summarizes exemplary data for children and minors in other scientific work (s. Table 3).

An Australian study presented the BLLs of 1 to 4-year-old children in the former lead-mining area of Broken Hill reported decreasing results over the years after remediation measures [16]. Geometric mean BLLs decreased from 163 µg/L in 1991 to 83 µg/L in 2007. However, mean BLLs remained substantially above the standard concentration of the general Australian population. The measures applied were health promotion, targeted clean-up, soil stabilization, as well as storm-water control.

A study performed in Peru on 1 to 6-year-old children in a mixed collective of children living in former or active mining sites reported a mean BLL of 72.0 µg/L [32], while in a mining town in Zambia, the mean BLLs were 197.0 µg/L in 6 to 12-year-old children [14]. A study conducted in the same region by Moonga et al. found hot spots with mean BLLs of 519 µg/L and cold spots with mean BLLs of 70 µg/L depending on the distance from the mining site and in relation to local prevailing winds [33]. In a study conducted on 9 to 12-year-old children in Mongolia, the mean BLLs were 74.2 µg/L [34]. That is, all of the studies reported mean BLLs that were 6 to 42 times higher than those measured in the present study.

A study providing long-term data from a former lead mining area in the US, focusing on 1 to 5-year-olds, reported mean BLLs that reached the mean BLL of the general population after rebatement measures. The mean BLLs in the mining area of Butte was 34.8 µg/L in 2003 and dropped to 15.3 µg/L in 2010, while a reference group of 1 to 5-year-old children decreased from 20.5 to 15.1 µg/L in the same time [15]. The mean BLL for eight German children from 2 to 5 years investigated in this study was 15.4 µg/L. This means they were in the same concentration range as in the former mining area of Butte after rebatement measures and the US general population values in 2010, respectively.

As mentioned earlier in the Method Section, seasonable effects on the BLL peaking in summer and autumn have been reported [4,26,27]. An explanation for the varying BLL contents between the locations cited in this discussion may therefore be found in the differing precipitations during the year and thus a pronounced effect of the predominating climate. While in Broken Hill in Australia, Corcona in Peru, Kabwe in Zambia, and Bayandalai in Mongolia, the climate is dryer, or in the case of Zambia at least t is affected by an extreme dry season, precipitations occur regularly, and in larger amounts in Butte in the USA and Mechernich and Kall in Germany. Stronger and frequent precipitations might effectuate a minor dusty air, and wash-out effects might occur, thus minimizing the possibility of respiratory and oral lead intake, resulting in overall reduced BLLs.

#### 4.4.2. Adults Living in the Vicinity of Mining Sites

Table 4 summarizes exemplary data for adults in other scientific work (s. Table 4).

In a Ghanaian study from a mining area with small-scale mining in Kenyasi performed with 18 to 57-year-old adults, a mean BLL of 50.0 µg/L was reported for occupational non-exposed workers, while a mean BLL of 68.0 µg/L was reported for small-scale miners [35].

In another study from Peru, 18 to 35-year-old non-miners living near a mining community, 4400 m above sea level showed a mean BLL of 47.5 µg/L, while unexposed persons in Lima had a mean BLL of 20.3 µg/L [36]. Furthermore, a mean BLL of 56.5 µg/L was reported for a group of people living near a mining area of Katanga in the Democratic Republic of Congo [37]. Overall, 106 or 116 µg/L, respectively, were reported from a study performed in Zambia on adult mothers and fathers of a mining town [14].

Finally, a study in South Korea presented a large amount of data on residents in the vicinity of former copper, gold, and silver mining areas. Unfortunately, lead mines were not included in the study [13]. However, since lead is often present in the aforementioned ores, lead and cadmium, blood levels were investigated. The 18 to 39-year-old residents had a mean BLL of 26.9 µg/L, 40 to 64-year-old residents had 32.3 µg/L, and residents above 65 years had 30.6 µg/L.

The observed mean BLLs were twice to seven times higher than the mean BLL reported in this study. Again, the influence of the climate might explain these findings as, apart from Korea, the reported mining sites tended to have dryer or extremer climates than that prevailing in the German region of Mechernich and Kall. The data from Korea, however, were the lowest among the ones cited. More research on the relationship between human metal absorption, the role of local or regional climate, and maybe the local vegetation would be a field worth further investigation in order to better appraise appropriate measures to reduce BLLs of the local population.

### 4.5. Policy Implications

In summary, the BLL observed in the region did not pose an immediate health threat and gave no hints that would require immediate action. Nevertheless, measures to reduce the lead burden should be undertaken, especially with regard to the children and adolescents in the region who had slightly elevated BLL in comparison to the general population since, following the state of today’s scientific knowledge about the toxicity of lead, no existing concentration of lead can be considered harmless. This is of special importance with regard to children.

Since the number of study participants between 3 and 17 years was too low to draw robust conclusions, the main recommendation is to perform a study with a statistical representative sample assessment of children and minors in the region.

From the data presented, several minor preventive and protective measures can be derived. In general, improved personal hygiene, e.g., more regular hand washing and avoiding hand-to-mouth contact, might reduce undesired lead ingestion. Other measures would comprise regular cleaning of the house to avoid the ingestion of lead-containing dust. People living in houses older than 1973 might also consider renovation measures, that is, the removal of potential lead-containing paints or plumbing. Overall, the local situation does not call for immediate action but for an awareness of the problem. Thus, regular information campaigns might help to maintain problem awareness.

## 5. Conclusions

Although the study was performed on a voluntary basis, the data allowed for not only the interpretation of the personal BLL but also provided information about the body burden of a segment of the adult population in the region. A comparison of the distribution of BLLs to existing reference values did not give a cause for concern in the adult population. RV exceedances occurred slightly more often than expected from the 95th percentile of the general population. Increasing BLLs related to age were observed, but since no RV exists for the population above 70 years, no further conclusions could be drawn. Where the RVs were exceeded, reductive measures should be clarified at any rate.

The data obtained for children and minors showed more cases of RV exceedances than expected. With the menace of adverse health effects already occurring at BLLs below 50 µg/L, measures to reduce the body burden are advised where the RVs were exceeded. Since the sample size of minors and children in the communities was small, no statements of a general health threat in the region could be derived. A statistical representative study is planned for children and adolescents to assess the local situation.

Among the factors investigated, occupancy and intense soil contact were identified as significant contributing factors that indicate an effect of soil lead on the BLL. The age of the house was also found to have a significant effect on the BLLs in the region. Surprisingly, other factors, such as the region of the real estate that was related to the soil lead concentration or the amount of home-grown food, did not turn out to contribute to the BLLs. Known contributing factors, such as age, sex, smoking, and the age of the property, were identified as contributors to BLLs.

Measures for lead exposure reduction comprise personal hygiene, especially in regard to children, and regularly cleaning the house. Older buildings should be checked for renovation measures in case of repeated exceedances of the RV. Regular information campaigns about the lead issue in communities should be considered to avoid exposure backlashes.

## Figures and Tables

**Figure 1 ijerph-19-06083-f001:**
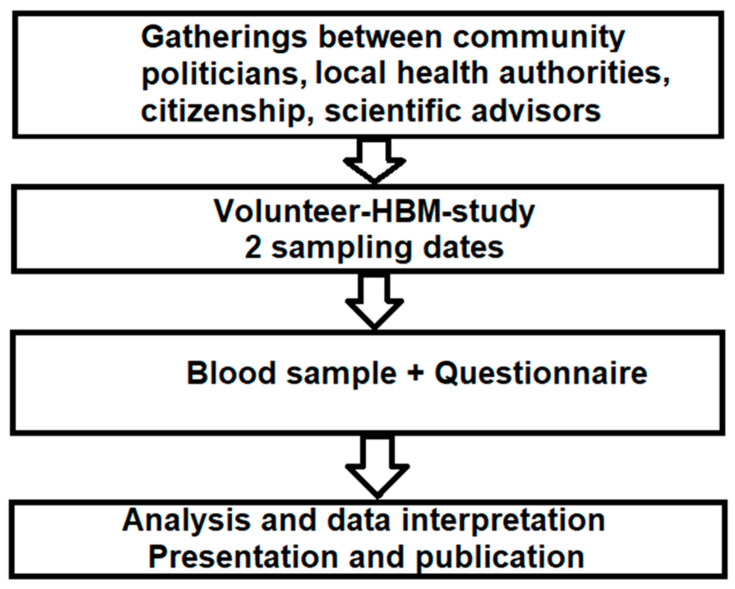
Conceptual framework of the Human Biomonitoring (HBM)-study conducted.

**Figure 2 ijerph-19-06083-f002:**
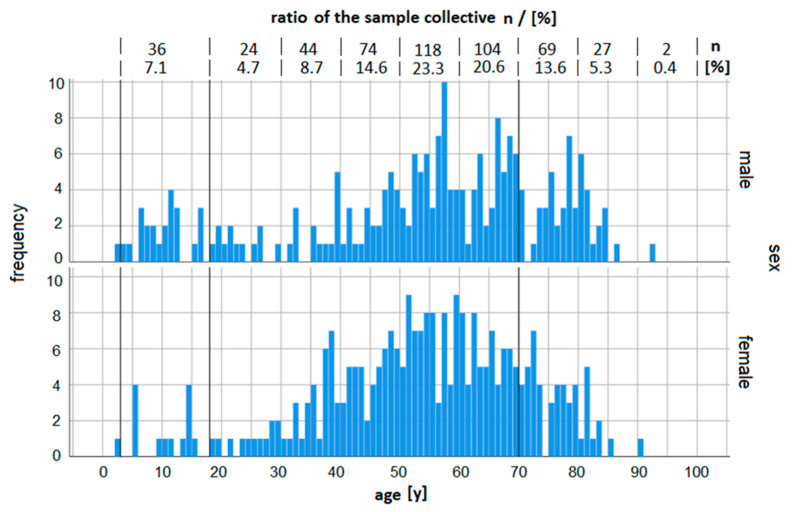
Age distribution of the participants depending on sex. Vertical lines indicate 3 years, 18 years, and 70 years. RV exist for 3 to 17-year-old persons and 18 to 70-year-old persons, only.

**Figure 3 ijerph-19-06083-f003:**
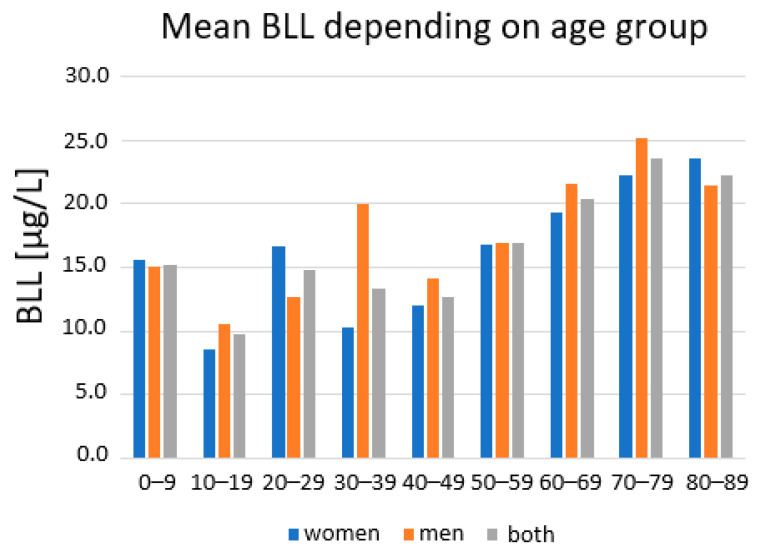
Mean BLL depending on sex and age of the volunteers.

**Figure 4 ijerph-19-06083-f004:**
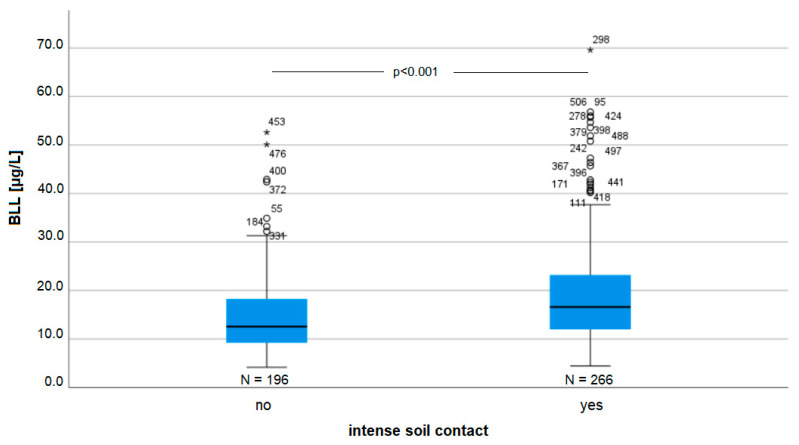
BLL and intense soil contact *p* = 0.035. Outliers marked with * are values above the third quartile plus 1.5 times the interquartile range.

**Figure 5 ijerph-19-06083-f005:**
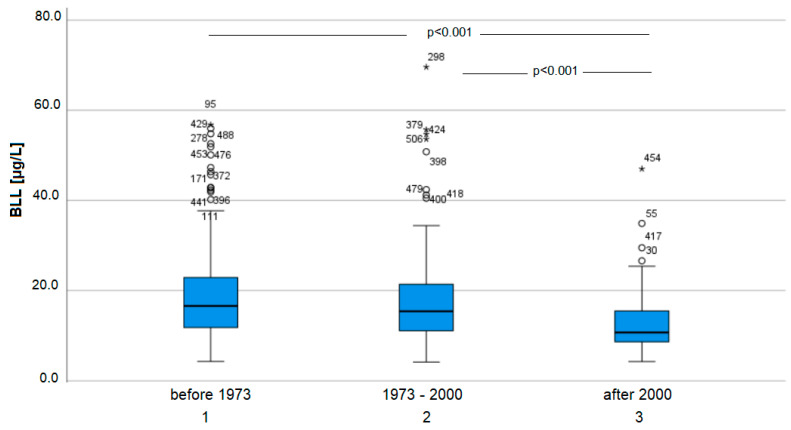
BLL and age of real estate *p* = 0.043. Outliers marked with * are values above the third quartile plus 1.5 times the interquartile range.

**Table 1 ijerph-19-06083-t001:** BLL results depending on sex and age compared to available German RVs [23]. * Single measurement. In brackets []: Non-existing RVs; Data compared to adult RVs.

				Pb µg/L				
	Age [y]	Amount	Min	Max	Mean	RV	Number > RV	>RV [%]
girls	<3	1	–	–	23.8 *	–	–	–
girls	3–17	13	4.3	25.4	10.6	15	2	15.4
boys	<3	1	–	–	12.4 *	–	–	–
boys	3–10	12	9.3	23.4	14.6	20	2	16.7
boys	11–17	11	6.2	26.6	10.8	15	2	18.2
women adults	18–69	216	4.2	56	15.4	30	14	6.5
men adults	18–69	148	5	69.6	17.8	40	9	6.1
women elderly	>70	49	4.5	54.8	22.2	–	[8]	[16.3]
men elderly	>70	49	9.3	55.7	24.4	–	[8]	[16.3]
unkown age & sex	–	6	12.5	25.4	17.7	–	–	–
total	–	506	4.2	69.6	17.4	–	–	–

**Table 2 ijerph-19-06083-t002:** Hierarchical regression analysis. Explanation of the BLL observed in the study population and significance.

	Standardized Regression Coefficiant Beta	*p*-Value	Variabel Type
age	0.218	0.000	numeric
occupancy	0.156	0.008	numeric
smoking	0.024	0.009	categorial
sex	−0.107	0.024	categorial
intense soil contact	0.104	0.035	categorial
age real estate	−0.102	0.043	numeric
garden time	0.095	0.058	numeric
region	−0.080	0.095	categorial
pregnancy	−0.042	0.374	categorial
bowels	−0.038	0.443	categorial

**Table 3 ijerph-19-06083-t003:** Comparison of our study to similar studies for children and minors.

					Mean		
Country	Region/Name	Sampling	Lead Source	Age [a]	BLL [µg/L]	n	Reference
Australia	Broken Hill	1991	Lead mine before remediation	1–4	163	n.d.	[16]
Australia	Broken Hill	2007	Lead mine after remediation	1–4	83	n.d.	[16]
Peru	Corcona, Tornamesa	2015	Several mining activities including lead; Past/Present	0–6	72	200	[32]
Zambia	Kabwe	2017	Lead-zinc mining town	6–12	197	208	[14]
USA	Butte	2003	Copper mining site	1–5	34.8	351	[15]
USA	Butte	2010	Copper mining site	1–5	15.3	461	[15]

**Table 4 ijerph-19-06083-t004:** Comparison of our study to similar studies for adults.

					Mean			
Country	Region/Name	Sampling	Lead Source	Age [a]	BLL [µg/L]	n	Reference	Country
Ghana	Kenyasi	2017	Gold mining area	18–57	50	40	[33]	Ghana
Peru	Cerro de Pasco	2012/2013	Mostly lead and zinc mining	18–35	47.5	157	[32]	Peru
Congo	Katanga	2017/2019	Copper and cobalt mining area	34–62	56.5	29	[35]	Congo
Zambia	Kabwe	2017	Lead-zinc mining town	adult mothers	106	404	[14]	Zambia
Zambia	Kabwe	2017	Lead-zinc mining town	adult fathers	116	125	[14]	Zambia
Korea	various	2008–2011	38 abandoned gold, silver, copper metal mine areas	18–39	26.9	164	[13]	Korea
Korea	various	2008–2011	38 abandoned gold, silver, copper metal mine areas	40–64	32.3	2077	[13]	Korea
Korea	various	2008–2011	38 abandoned gold, silver, copper metal mine areas	>65	30.6	3441	[13]	Korea

## Data Availability

Not applicable.

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
