# Peer review of "Blood Lead Monitoring in a Former Mining Area in Euskirchen, Germany—Volunteers across the Entire Population"

_ijerph, 2022, doi:10.3390/ijerph19106083_

Round 1

Reviewer 1 Report

General comments:

The manuscript addresses the issue of lead impregnation in the population in a mining area where historical soil contamination is known. The article is well structured and easy to read. While the study is not particularly original, the results are of undeniable interest for the management of this situation and for sites with similar characteristics. The main methodological limitation, in my opinion, is the voluntary recruitment of participants, which may introduce a bias. This methodological choice, probably based on practical considerations, is however well explained by the authors. 

Specific comments 

  • The introduction should present in more detail the toxicokinetics of lead and in particular its elimination kinetics (e.g., lead related to recent incorporation vs. release from bone). This would probably allow for a better discussion of the interpretation of the results with respect to current exposure and historical contamination.
  • Table 1. It is mentioned that maximum values are found for 5 year old girls (25.4 ug/L) and 11 year old boys (26.6 ug/L). These are obviously maxima relative to the German reference value and not absolute values. It should be clarified which endpoint (metric) is used.
  • Table 1. The reference value for boys aged 11-17 years is lower than the reference value for boys aged 3-10 years. Is this correct?
  • Figure 3. For some age groups, there is a marked difference between the contamination in males and females. The authors should comment on these differences (is this occupational exposure)?

Author Response

Dear reviewer,

thank you very much for you comments on the draft. Please find the replies to the topics raised below. We hope to have met your points.

With kind regards

Jens Bertram

Reply to Reviewer 1

Reviewer 1 wrote:

“The introduction should present in more detail the toxicokinetics of lead and in particular its elimination kinetics (e.g., lead related to recent incorporation vs. release from bone). This would probably allow for a better discussion of the interpretation of the results with respect to current exposure and historical contamination.”

Reply to Reviewer 1: We added in line 31ff:

“Lead binds primarily to hemoglobin and is therefore easily distributed throughout the body. Excretion takes place via urine and feces. Half-life time of lead in blood is about 30 days. The BLL reflects the exposure of the previous months. Bone and teeth are depots for long-term storage, with approximately 94 and 76% of the body burden deposited in the bone in adults or children respectively, thus representing the cumulative long-term exposure.”

And in the discussion section line 259:

“These findings can be explained with increased lead release from decreasing bone substance occurring as part of the aging process of adults, maybe resembling the overall, ten-fold higher lead burden during the decades when leaded fuels were used.”

Reviewer 1 wrote:

“Table 1. It is mentioned that maximum values are found for 5 year old girls (25.4 µg/L] and 11 year old boys (26.6 µg/). These are obviously maxima relative to the German reference value and not absolute values. It should be mentioned which endpoint (metric) is used”

Reply to Reviewer 1:

As stated in the text in line 160f these were the two study subjects (“a 5-year-old girl” and “an 11-year-old boy”) with the highest BLL for underage girls or boys that we measured in the study at hand. To clarify that these were maxima observed in our study we added in line 158:

“..in our investigation”

Reviewer 1 wrote:

“The reference value for boys aged 11-17 years is lower than the reference vale for boys aged 3-10 years. Is this correct?”

Reply to Reviewer 1:

The reference value for boys aged 11-17 years (15 µg/L) is indeed lower, than for boys aged 3-10 years (20µg/L).

Reviewer 1 wrote:

“Figure 3. For some age groups, there is a marked difference between the contamination in males and females. The author should comment on these differences (is this occupational exposure)?”

Reply to Reviewer 1: We added in lines 175ff:

“In the age groups of the 20 to 29-year-olds and the 30 to 39-year-olds differences between both sexes can be observed. These variances might be explained with 5 individuals that showed marked RV exceedances in comparatively small subgroups (s. Figure 2.), thus having a major effect of the mean BLL. 3 of these 5 reported intense soil contact while one did not make a statement. No professional contact was stated.”

Reviewer 2 Report

This is a relevant and interesting survey of blood lead levels in an area of Germany that has a long history of mining.  The significance of the results is constrained by the limited number of subjects, particularly amongst children, who are most affected by lead exposure.  Still, the results showed an interesting pattern, with blood lead levels relatively high in young children (<9 years) then dropping off and, then generally gradually rising with age.  It would be interesting to try learn more about lead exposures in this area.  Was there a period of time when older people were exposed to especially high levels of lead, beyond leaded gasoline & paint?  Was farming/gardening more common in the past? Or do the higher BLL in older people simply reflect gradual increase in the body burden over time?  It appears that there is not a lot of data looking at lead levels in historic mining sites across the life course, but a little more discussion of these questions would be valuable.

I suggest that you make a stronger argument for the significance of your findings by emphasizing that there does not appear to be any safe level of lead in the blood, as asserted by the WHO and other organizations.  Your finding of blood lead levels of 15 micrograms/L in young children is notable and calls for strategies to reduce exposure.  As you suggested, further study of this group will yield important information. 

Author Response

Dear reviewer,

thank you very much for you comments on the draft. Please find the replies to the topics raised below. We hope to have met your points.

With kind regards

Jens Bertram

Reply to Reviewer 2

Reviewer 2 wrote:

“It would be interesting to learn more about lead exposures in the area. Was there a period of time when older people were exposed to especially high levels of lead, beyond leaded gasoline and & paint? Was farming/gardening more common in the past? Or do the higher BLL in older people simply reflect gradual increase in the body burden over time?”

Reply to Reviewer 2:

We added in lines 258ff of the discussion:

“These findings can be explained with increased lead release from decreasing bone substance occurring as part of the aging process of adults, most likely resembling the overall, ten-fold higher lead burden during the decades when leaded fuels were used. To the authors knowledge other special activities, like the amount of farming activities in the region did not change much in the area since the closure of the mine in 1957 and therefore does not explain the trend to higher BLL in elder people.”

Reviewer 2 wrote:

“I suggest that you make a stronger argument for the significance of your findings by emphasizing that there does not appear to be any safe level of lead in blood,…”

Reply to Reviewer 2: We added in line 36f and in lines 398ff

“….To date no BLL that can be considered as safe exists.”

“…since, following the state of todays´ scientific knowledge about the toxicity of lead, no concentration of lead that can be considered as harmless exists. This is of special importance with regard to the children.”

Reviewer 3 Report

The manuscript is well-designed and well-written. I congratulate the authors on your interesting paper. My minor points are presented below:

1) Material and methods, 2.1: I would write on which anticoagulant blood sample was taken.

2) Line 282: there is “hat”, it should be “have”

3) Line 309: “summarized in table 3 for children and minors and table 3 for adults” Table 4 present data for adults. Moreover, I would put table 3 and table 4 just below the paragraph in which you mentioned these tables.

Author Response

Dear reviewer,

thank you very much for you comments on the draft. Please find the replies to the topics raised below. We hope to have met your points.

With kind regards

Jens Bertram

Reply to Reviewer 3

Reviewer 3 wrote:

“1) Material and methods, 2.1: I would write on which anticoagulant blood sample was taken.”

Reply to Reviewer 3: We added in the according paragraph:

Line 105: “Blood samples were collected on 4.9 mL EDTA monovettes (Sarstedt, Nümbrecht, Germany) and stored in a fridge overnight”

Reviewer 3 wrote:

“2) Line 282: there is “hat”, it should be “have””

Reply to Reviewer 3:

Line 300: We corrected “hat” to “have”

Reviewer 3 wrote:

“3) Line 309: “summarized in table 3 for children and minors and table 3 for adults” Table 4 present data for adults. Moreover, I would put table 3 and table 4 just below the paragraph in which you mentioned these tables.”

Reply to Reviewer 3:  We deleted in line 326 “and is summarized in table 3 for children and minors in table for adults.”

Instead we added in line 329f “Table 3 summarizes exemplary data for children and minors in other scientific work (s.Table 3).”

We added in line 365 chapter 4.4.2: “Table 4 summarizes exemplary data for adults in other scientific work (s. Table 4).” And moved Table 4 accordingly beneath.

We further noted an inconsistency in our citation of Tables and Figures and changed them from the abbreviated versions (Fig. or Tab.) to the full term (Figure or Table).

We further spell checked the manuscript and corrected Line 11 from “concentration of soil” to “concentration in soil”

The numbering in the affiliation was redundant. We therefore deleted the numbering in line 5 and 8.